# Neoadjuvant Immunotherapy for Patients with dMMR/MSI-High Gastrointestinal Cancers: A Changing Paradigm

**DOI:** 10.3390/cancers15153833

**Published:** 2023-07-28

**Authors:** Muhammet Ozer, Charan Thej Reddy Vegivinti, Masood Syed, Morgan E. Ferrell, Cyndi Gonzalez Gomez, Svea Cheng, Jennifer Holder-Murray, Tullia Bruno, Anwaar Saeed, Ibrahim Halil Sahin

**Affiliations:** 1Department of Gastrointestinal Oncology, Dana Farber Cancer Institute, Boston, MA 02215, USA; 2Department of Medicine, Jacobi Medical Center, Albert Einstein College of Medicine, Bronx, NY 10461, USA; 3Department of Medicine, University of Pittsburg School of Medicine, Pittsburgh, PA 15213, USA; 4Department of Surgery, University of Pittsburgh School of Medicine, Pittsburgh, PA 15213, USA; 5Department of Immunology, University of Pittsburgh School of Medicine, Pittsburgh, PA 15213, USA; 6Division of Hematology/Oncology, Department of Medicine, University of Pittsburgh School of Medicine, Pittsburgh, PA 15213, USA

**Keywords:** immunotherapy, neoadjuvant therapy, gastrointestinal cancers, immune checkpoint inhibitors

## Abstract

**Simple Summary:**

Immune checkpoint inhibitors (ICIs) have transformed the treatment of mismatch repair-deficient (MMR-D)/microsatellite instability-high (MSI-H) colorectal cancers. These cancers have a high mutation burden and generate antigens called mutation-associated neoantigens (MANAs). ICIs activate T-cell immunity against these cancers, resulting in significant antitumor activity. Dramatic responses seen in patients with advanced-stage MSI-H gastrointestinal cancers have resulted in rapid clinical development of ICIs for patients with earlier stages of disease. Based on the available evidence, the pathological response observed in this context shows great promise, which encourages the use of ICIs as neoadjuvant therapy.

**Abstract:**

Immune checkpoint inhibitors have revolutionized the management of mismatch repair-deficient (MMR-D)/microsatellite instability-high (MSI-H) gastrointestinal cancers, particularly colorectal cancer. Cancers with the MMR-D/MSI-H genotype often carry a higher tumor mutation burden with frameshift alterations, leading to increased mutation-associated neoantigen (MANA) generation. The dramatic response seen with immune checkpoint inhibitors (ICIs), which are orchestrated by MANA-primed effector T cells, resulted in the rapid development of these novel therapeutics within the landscape of MSI-H gastrointestinal cancers. Recently, several clinical trials have utilized ICIs as potential neoadjuvant therapies for MSI-H gastrointestinal cancers and demonstrated deep clinical and pathological responses, creating opportunities for organ preservation. However, there are potential challenges to the neoadjuvant use of ICIs for certain disease types due to the clinical risk of overtreatment for a disease that can be cured through a surgery-only approach. In this review article, we discuss neoadjuvant management approaches with ICI therapy for patients with MSI-H gastrointestinal cancers, including those with oligometastatic disease. We also elaborate on potential challenges and opportunities for the neoadjuvant utilization of ICIs and provide further insight into the changing treatment paradigm of MMR-D/MSI-H gastrointestinal cancers.

## 1. Introduction

The increases in industrialization and an aging population have played a significant role in the increasing incidence and mortality rates of gastrointestinal cancers. Colorectal cancer (CRC) is currently the leading gastrointestinal cancer in Western countries. CRC is ranked third in incidence and second in mortality rate among all cancers, representing a major health burden in Western countries [1]. Meanwhile, gastric cancer is responsible for over one million new cases and 700,000 deaths annually worldwide. Gastric cancer is the fifth-most-diagnosed cancer and the fourth-most-common cause of cancer-related deaths globally and is more commonly seen in Asian countries [1]. With the advancements in surgical techniques and systemic therapies, the outcomes have improved for most gastrointestinal cancers over the last decade. As molecular genotyping has significantly evolved due to technical advancements in next-generation sequencing (NGS) determining biological signatures, the treatment algorithms for several solid tumors have significantly changed with more precise therapies, including immune checkpoint inhibitors [2].

The emergence of molecular profiling technology allowed clinicians to examine microsatellite instability in several solid tumors [3]. The mismatch repair (MMR) system consists of specific DNA mismatch repair enzymes and depends on four essential genes: mutL homolog 1 (MLH1), postmeiotic segregation increased 2 (PMS2), mutS homolog 2 (MSH2), and mutS 6 (MSH6) [4]. The inactivation of any MMR enzyme can cause dysfunctional or unexpressed proteins, defined as deficient MMR (dMMR), and induce insertion/deletion alterations, particularly in microsatellite regions, leading to microsatellite instability (MSI). MSI-associated frameshift alterations result in the production of significant mutation-associated neoantigens (MANAs), which represent unique characteristics of MSI-H cancers [5,6]; see Figure 1. Cancers with dMMR are considered to be MSI-high (MSI-H); therefore, in the literature, the terms dMMR and MSI-H are used interchangeably; however, even though it is rare, dMMR tumors may represent MSI-low disease or even microsatellite-stable (MSS) diseases [7,8]. dMMR can be due to germline MMR gene mutations; more commonly, it is linked to sporadic events, including promoter hypermethylation [9]. Notably, while dMMR/MSI-H colon cancer mostly occurs as a result of somatic events in MMR genes and, less commonly, as a result of germline mutations, dMMR/MSI-H rectal cancers are more likely to be linked to Lynch Syndrome [10].

The dMMR/MSI-H disease is seen in various gastrointestinal cancers, most commonly CRC and small bowel cancer [11,12]. dMMR/MSI-H CRC accounts for 12–20% of patients with colon cancer and is more common in stage I-III CRC [13,14]. Approximately 5% of rectal cancer patients can present with dMMR/MSI-H disease [15]. Meanwhile, 4–8% of gastric and gastroesophageal junction (GEJ) cancers and ~20% of small bowel adenocarcinomas are linked to dMMR/MSI-H [16,17,18]. Notably, numerous studies have demonstrated that dMMR/MSI-H cancers lack benefit from adjuvant and/or neoadjuvant fluoropyrimidine, which is the backbone agent of chemotherapeutic regimens used in colorectal cancer [19,20,21,22,23,24]. The promising response seen in advanced-stage MSI-H gastrointestinal cancers [25] triggered clinical trials utilizing ICIs as neoadjuvant therapies for early-stage dMMR/MSI-H CRC and gastric and GEJ cancers and revealed impressive clinical results [26,27,28,29,30]. This review article provides an overview of recent advancements in neoadjuvant immunotherapy for patients with dMMR/MSI-H gastrointestinal cancers. We also elaborate on the challenges and advantages of neoadjuvant ICI therapy and attempt to shed light on future therapeutic opportunities in the neoadjuvant space for locally advanced and oligometastatic MSI-H gastrointestinal cancers.

## 2. Neoadjuvant Immunotherapy for Patients with dMMR/MSI-H Rectal Cancer

The standard therapies for locally advanced rectal cancer remain surgery, chemoradiation, and systemic chemotherapy. The current literature supports that dMMR/MSI-H rectal cancer is inherently resistant to fluoropyrimidine, leading to unfavorable survival outcomes [20,21,31,32]. Notably, surgical resection and/or radiotherapy carries a significant risk for ostomy creation, fecal incontinence, and/or urinary and sexual dysfunction [33,34,35]. Therefore, over the last few decades, investigators have focused on a total neoadjuvant approach in locally advanced rectal cancer treatment to provide organ preservation and improve long-term prognosis [36,37,38,39]. Early studies with total neoadjuvant trials showed that patients with rectal cancer harboring MMR-D undergoing induction chemotherapy might experience disease progression, while this was not noted for patients with MSS disease [21]. Therefore, it has become evident that a total neoadjuvant approach with induction chemotherapy may not be the ideal therapeutic approach for patients with MMR-D/MSI-H rectal cancer.

ICI therapies have been approved for patients with metastatic dMMR/MSI-H CRC as first-line therapies with durable responses [40]. Fluoropyrimidine resistance seen with MSI-H biology and the breakthrough efficacy of ICI therapy in the metastatic setting led to many clinical trials to evaluate the neoadjuvant role of these agents for patients with locally advanced dMMR/MSI-H rectal cancer. Cercek et al. conducted a single-institution phase II trial with dostarlimab monotherapy, an anti-PD1 monoclonal antibody, in locally advanced stage II or III dMMR/MSI-H rectal cancer [28]. Patients received dostarlimab 500 mg every three weeks for approximately six months. Per the study design, patients with complete clinical response (CR) would proceed with a watch-and-wait approach, while those with incomplete responses were planned to undergo salvage chemoradiation and surgery. In this study, a total of 12 patients completed the dostarlimab induction therapy with a minimal follow-up of 6 months (Table 1). Notably, all 12 patients (100%) achieved clinical CR consistent with the promising efficacy of immune checkpoint inhibitors, and none of the patients required chemoradiation or surgery. The most common immune-related adverse events were dermatitis, thyroid disorders, and asthenia. Notably, there were no adverse events beyond grade 2. Pembrolizumab monotherapy was also investigated in a phase II open-label, single-center trial of locally advanced dMMR/MSI-H solid tumors, including 8 rectal and 19 colon cancer patients [41]. The best ORR was 82%, and the pathologic CR was 79% among CRC patients who underwent surgical resection. There were no new safety signals reported. Wang et al. evaluated ten patients with locally advanced dMMR/MSI-H rectal cancer who achieved complete clinical CR after receiving neoadjuvant anti-PD-1 immunotherapy and adopted the watch-and-wait strategy. No local or distant relapse was seen during a median follow-up of 11.6 months [42]. A retrospective case series study investigated neoadjuvant single-agent PD-1 inhibitors in eight locally advanced dMMR/MSI-H rectal cancer patients [43]. Three patients (37.5%) achieved clinical CR and a watch-and-wait strategy was adopted for these patients. In the same study, among patients who underwent surgical resection for either MSI-H colon or rectal cancer, pathological CR was 75%, indicating there is a significant role in organ preservation with immune checkpoint inhibitor therapy for patients with locally advanced MSI-H rectal cancer.

The sequential use of ICIs with chemoradiation was also investigated in several studies. Bando et al. investigated neoadjuvant nivolumab plus chemoradiotherapy in patients with locally advanced dMMR/MSI-H and pMMR/MSS rectal cancer [44]. Three of five dMMR/MSI-H rectal cancer patients achieved pathologic CR (60%). Perhaps it is important to note that radiation may directly affect the effectiveness of tumor-infiltrating lymphocytes, which are key immune cells to mediate ICI response. A phase II VOLTAGE-A trial evaluated the use of five cycles of nivolumab after chemoradiation in locally advanced dMMR/MSI-H and pMMR/MSS rectal cancer [45]. At the time of the preliminary results of this trial, 5 patients with MSI-H rectal cancer were enrolled, and pathologic CR was noted to be 60% (3/5). None of the patients with MSI-H rectal cancer had recurrent disease. Currently, the ECOG-ACRIN phase II trial (EA2201) is actively investigating whether neoadjuvant nivolumab plus ipilimumab and short-course radiation in locally advanced dMMR/MSI-H rectal cancer would result in increased pCR rates (NCT04751370). However, this study has been modified due to the highly promising outcomes of dostarlimab monotherapy, and patients will undergo interim evaluation following immunotherapy induction therapy, and patients who achieve complete clinical response will undergo a watch-and-wait approach without a short course of radiation. Another phase I study investigating pembrolizumab in combination with conventional chemoradiation with capecitabine is actively recruiting (NCT04357587). The accumulating evidence suggests a majority of patients with MSI-H rectal cancer may not need chemoradiation, and long-term adverse complications of pelvic radiation should be considered when investigating ICI therapy in this setting.

**Table 1 cancers-15-03833-t001:** Clinical studies investigating immune checkpoint inhibitors as neoadjuvant therapy for MSI-H gastrointestinal cancers.

Trial Name	Drug	Cancer	Design	MSI-H (N)	RR (%)
Andre et al. [26]	Nivolumab and ipilimumab	Localized esophageal and gastric adenocarcinoma	Phase II, prospective, single-arm, open-label	32	A total of 91% (29/32) of patients underwent surgery. A total of 59% (17/29) of patients who underwent surgery had pCR.
Chalabi et al. (NICHE-1) [27]	Nivolumab and ipilimumab	Colon cancer	Phase II, prospective single-arm, open-label	20	A total of 100% (20/20) had PR, of which 95% (19/20) had MPR, and 60% (12/20) had pCR.
Chalabi et al. (NIHCE-2) [46]	Nivolumab and ipilimumab	Colon cancer	Phase II, prospective single-arm, open-label	112	A total of 93% (102/107) had MPR and 67% (72/107) achieved pCR.
Cercek et al. [28]	Dostarlimab	Rectal cancer	Phase II, prospective, single-arm	12	A total of 100% achieved cCR.
Ludford et al. [41]	Pembrolizumab	Colorectal cancer	Phase II open-label, single-center trial	35 (27—CRC patients, 8—non-CRC patients)	Pathological response: A total of 49% (17/35) underwent surgery, of which 14 had CRC; of these, 79% (11/14) had pCR. Radiographic response—RR-82% (27/33), CR—30% (10/33), PRs-52% (17/33).
Wang et al. [42]	Anti-PD-1 immunotherapy	Rectal cancer	NR	10	One-year local recurrence-free survival, distant metastasis-free survival, and disease-free survival for the whole cohort were 100%, 100%, and 100%.
Zhang et al. [43]	Anti-PD-1 immunotherapy (4 patients—pembrolizumab, 9 patients—sintilimab, 19 patients—tiselizumab)	Colorectal cancer	Retrospective, single-center, case series study	32 (8 patients—LARC, 24 patients—LACC)	Three LARC patients achieved cCR and W&W strategy was adopted. RR among 29 patients with radical surgery was 100% (29/29), the PR was 100% (29/29), the MPR was 86.2% (25/29), and pCR was 75.9% (22/29).
Barraud et al. [47]	A total of 34 (76%) received monotherapy with anti-PD1 or anti-PDL1 monoclonal antibodies, and 11 (24%) a combination of anti-PD1 plus anti-CTLA4 monoclonal antibodies	Colorectal cancer with isolated peritoneal carcinomatosis	Multicentric cohort study	45	RR—46% with cCR in 24% (11/45) of patients and partial responses in 22% (10/45) of patients. OS and PFS were not reached. A total of 18% (8/45) of patients underwent surgery, of which 88% (7/8) of patients had pCR.

## 3. Neoadjuvant Immunotherapy for Patients with dMMR/MSI-H Colon Cancer

Similar to rectal cancer, studies have investigated the role of immune checkpoint inhibitor therapy as neoadjuvant therapy for patients with MSI-H colon cancer. The NICHE study pioneered the use of neoadjuvant ICI therapy for stage I (10%), II (10%), and III (80%) dMMR/MSI-H colon cancer [27]. The study investigated neoadjuvant nivolumab (anti-PD-1 antibody) plus ipilimumab (anti-CTLA-4 antibody) in resectable colon cancer patients with either dMMR/MSI-H or pMMR/MSS. Patients received two doses of 3 mg/kg nivolumab and a single dose of 1 mg/kg ipilimumab. All patients with dMMR/MSI-H colon cancer (32/32) achieved a pathologic response, and among those, 97% (31/32) of them had a major pathologic response (less than 10% viable tumor), and 22/32 (69%) of them achieved complete pathologic response after a median follow-up time of 25 months. The nivolumab plus ipilimumab combination was safe, feasible, and well tolerated. Subsequently, the NICHE-2 study evaluated the nivolumab plus ipilimumab combination in a larger cohort of 107 nonmetastatic dMMR/MSI-H colon cancer patients [46]. A major pathologic response was observed in 102 patients (95%); among those, 72 patients had pathological CR (67%), indicating a deep pathological response with neoadjuvant immunotherapy. At a median follow-up of 13 months, none of the patients had disease recurrence. Immune-related severe adverse events (grade 3 or 4) were seen in only three patients (3%). The PICC study, a phase II trial, is currently investigating neoadjuvant toripalimab (anti-PD1 monoclonal antibody) with or without celecoxib in 34 patients with locally advanced dMMR/MSI-H colon cancer [48]. The patients were treated for up to 6 months, followed by surgery. Early results showed that pathologic CR rates were 88% and 65% with the toripalimab plus celecoxib and toripalimab-only groups, respectively. At a median follow-up of 14.9 months, all patients were alive and disease-free, consistent with a highly effective approach with neoadjuvant immunotherapy.

Zhou et al. published a meta-analysis of neoadjuvant immunotherapy for nonmetastatic CRC. Among 113 cases with nonmetastatic dMMR/MSI-H CRC, the pathologic CR and major pathologic response were 63.9% and 80.3%, respectively [49]. Another phase II study is currently evaluating neoadjuvant tislelizumab (anti-PD1 antibody) in locally advanced dMMR/MSI-H CRC (NCT05116085). Early results reported that all patients (18/18, 100%) achieved significant regression. Five patients with locally advanced dMMR/MSI-H colon cancer had confirmed pathologic CR after surgery [13]. The NEOPRISM-CRC trial is also currently investigating pembrolizumab in patients with dMMR/MSI-H CRC or high TMB (≥20 mutations per megabase) [50].

Although promising clinical effects of neoadjuvant immunotherapy in locally advanced dMMR/MSI-H CRC have been reported, there are no controlled clinical trials to compare adjuvant ICI therapy with the neoadjuvant approach [51,52]. In the ATOMIC trial, patients with MSI-H stage III colon cancer were randomized into chemotherapy vs. chemoimmunotherapy with atezolizumab and FOLFOX to evaluate the additive role of immune checkpoint therapy in the adjuvant setting. However, atezolizumab was not investigated as a monotherapy in this study, and whether these patients even need chemotherapy will not be answered in this clinical trial. More studies comparing neoadjuvant and adjuvant immune checkpoint inhibitor therapies should be investigated to better define the role of neoadjuvant immunotherapy for locally advanced colon cancer.

## 4. Neoadjuvant Immunotherapy for Patients with dMMR/MSI-H CRC and Distant Metastasis

The effectiveness of ICI therapy for advanced-stage CRC has been well established; however, little is known about the depth of pathologic response for those undergoing metastasectomy or cytoreductive surgery. Jin and colleagues conducted a retrospective controlled study with 75 dMMR/MSI-H metastatic CRC patients [53]. Among them, 16 patients had metastasectomy of liver lesions. Six patients received adjuvant pembrolizumab. The median overall survival (OS) was significantly higher among the dMMR/MSI-H metastasectomy group compared with those who did not undergo metastasectomy (82 mo vs. 13.9 mo, retrospectively). Also, it was evident that patients with dMMR/MSI-H metastatic CRC had a greater benefit from metastasectomy compared with the pMMR/MSS group (median OS: 82 mo vs. 69.9 mo, retrospectively). However, this study did not include patients with neoadjuvant ICI therapy; therefore, the role of ICI therapy remains unclear for patients with resectable liver metastasis yet should be considered, given the deep and durable responses seen with ICIs. Whether pathological response impacts the long-term outcomes and whether metastasectomy is even needed in this population warrant future studies. Notably, several case series and reports of CRC patients with MSI-H CRC with peritoneal metastasis have shown promising results with ICI therapy. In a case series of patients with MSI-H CRC with peritoneal carcinomatosis, eight patients underwent cytoreductive surgery after induction ICI therapy, and notably, seven out of eight patients (87%) achieved complete pathological response [47]. This dramatic pathologic response was also noted in other case reports. Tonello et al. reported a patient with Lynch Syndrome who was treated with nivolumab 240 mg every two weeks. After two years of stable disease with ICI therapy, the patient underwent cytoreductive surgery (CRS), which showed a complete pathologic response [54]. After nine months of follow-up, no disease recurrence was noted. In another case report, the authors discussed the case of a 46-year-old patient with MSI-H CRC with peritoneal carcinomatosis without a history of Lynch presented who was treated with pembrolizumab monotherapy. The patient achieved a complete pathological response confirmed via tissue samples obtained during the cytoreductive surgery [55]. These results, although limited, indicate that deep pathological responses can be seen among patients with MSI-H CRC and peritoneal carcinomatosis. Further large-scale studies are needed for tailoring surgical strategies, including colorectal resection, metastasectomy, debulking, HIPEC, and/or the watch-and-wait approach for patients with dMMR/MSI-H metastatic CRC in the immunotherapy era.

## 5. Neoadjuvant Immunotherapy for Patients with dMMR/MSI-H Gastric and Esophagogastric Junction Cancers

Over the past few decades, research has been directed towards integrating systemic treatments for locally advanced gastric and GEJ cancers to achieve better survival outcomes and increase the R0 surgical resection rates [56,57,58,59]. The current standard of care for patients with locally advanced gastric cancer consists of neoadjuvant chemotherapy or surgical resection followed by adjuvant chemotherapy [56,60,61]. The MAGIC trial showed a 79% R0 resection rate for neoadjuvant chemotherapy in locally advanced gastric and GEJ cancers [56]. Despite this, the pathologic response association with OS was suboptimal, with approximately 10% pathologic CR. A meta-analysis of four randomized clinical trials evaluated the prognostic value of MSI in gastric cancer patients [17]. The authors reported that dMMR/MSI-H was associated with better OS and disease-free survival (DFS) at five years. However, patients with gastric cancer with dMMR/MSI-H had worse outcomes when treated with perioperative chemotherapy [62]. Furthermore, neoadjuvant chemotherapy led to an increased incidence of complications related to surgery [63].

The dMMR/MSI-H phenotype is seen in approximately 5–10% of gastric and GEJ cancers [16,17,18]. As noted in several studies, dMMR/MSI-H is associated with a decreased benefit from fluoropyrimidine-based chemotherapy with respect to OS and DFS [17,64]. In Checkmate 649, MSI-H was the most important predictor of response to chemoimmunotherapy, consistent with the observations seen among patients with MSI-H CRC [65]. Due to these facts, studies have investigated ICI therapy as a neoadjuvant treatment. In a randomized, open-label, phase II trial (DANTE), Al-Batran et al. reported improved pathologic CR by adding atezolizumab to neoadjuvant platinum and fluoropyrimidine-based chemotherapy in patients with resectable GEJ cancer, particularly those with MSI-H disease [66]. In the phase II NEONIPIGA trial, Andre et al. evaluated a neoadjuvant nivolumab plus ipilimumab combination in locally advanced dMMR/MSI-H gastric or GEJ cancer patients [26]. In the study, 50% (16 patients) had gastric cancer, and the other half had GEJ cancer. Patients received nivolumab 240 mg once every two weeks for six cycles and ipilimumab 1 mg/kg once every six weeks for two cycles, followed by surgery and adjuvant nivolumab 480 mg once every four weeks for nine cycles. Three patients had a complete CR and did not have surgery. All 29 patients who underwent surgery had R0 resection, and 58.6% (17 patients) had pathologic CR. At 12 months of median follow-up, 93.7% (30 patients) were alive without recurrence. The nivolumab plus ipilimumab combination treatment was well tolerated, with grade ≥3 immune-related adverse events of 25%. Currently, a multicenter, phase II INFINITY trial (NCT04817826) is investigating the efficacy and safety of a neoadjuvant tremelimumab plus durvalumab combination in dMMR/MSI-H gastric or GEJ cancer patients [67]. In the first cohort of this trial, 15 patients were evaluated after a single dose of tremelimumab and three doses of durvalumab once every four weeks, followed by surgery. The pathologic CR was 60% (9/15), and the major complete pathologic response was 80%, indicating that deep pathological response may offer a path for a potential nonoperative approach for patients with MSI-H gastric and GEJ cancers. Grade ≥3 immune-related adverse events were observed in three patients; all responded well to high-dose steroids [68].

In a meta-analysis, Li et al. assessed 21 prospective phase I/II studies comprising 687 patients with locally advanced gastric or GEJ cancer treated with neoadjuvant ICI therapy [69]. They reported that dMMR/MSI-H was associated with improved pathological CR and major pathologic response rates than pMMR/MSS when patients are treated with ICI therapy. Despite their heterogeneity, the results suggest that ICI-based neoadjuvant treatment is safe and feasible and produces superior pathologic responses for patients with the dMMR/MSI-H phenotype.

## 6. Discussion

ICIs have dramatically changed the treatment paradigm of several tumors within the last decade. Most recently, clinical studies indicated that neoadjuvant ICI therapy significantly benefits patients with dMMR/MSI-H CRC, gastric, and GEJ cancers. Although no established predictive biomarker exists for treatment response for CRC and gastric cancers, MSI appears predictive and reliable in estimating 5FU resistance, particularly for patients with CRC and inferior response to neoadjuvant chemotherapy. Consistent with this, several studies reported low pathologic CR observed among patients with dMMR/MSI-H GI cancers treated with fluoropyrimidine-based neoadjuvant chemotherapy [21,32,60,70,71]. It is important to recognize that the tumor microenvironment of MSI-H CRC is highly different than its MSS counterpart, mainly due to MANA-primed T-cell infiltration. Several studies have shown that MSI-H CRC is highly enriched with effector T cells in the tumor microenvironment and in the blood circulation, and these cells facilitate antitumor response augmented by ICIs [72]. Therefore, it is essential to test all patients with CRC, gastric, and GEJ cancers for MSI/MMR to better define a therapeutic approach in light of well-established fluoropyrimidine resistance and ICI-responsive biology [22,31]. Several studies reported that ICIs are more beneficial when utilized in earlier lines of treatment compared with their use for advanced-stage disease [40,73].

At this juncture, the most important question that remains to be determined is the identification of appropriate primary endpoints for clinical trials investigating neoadjuvant ICI therapies. It is important to note that primary endpoints may very well differ based on the primary site of cancer. For example, for rectal cancer, the management has shifted to organ preservation, given the high incidence of lifelong colostomy requirement with surgical intervention. Therefore, studies with MSI-H rectal cancer should primarily aim to achieve organ preservation while maintaining DFS and OS outcomes. However, it is unclear whether organ preservation would be the ideal primary endpoint for colon cancer, in which organ preservation is relatively less clinically relevant as the vast majority of patients do not require permanent colostomy. Moreover, organ preservation for right-sided colon cancer brings potentially more challenging clinical scenarios where patients may need frequent colonoscopies, and it is yet not well studied whether imaging modalities can assess local disease with reasonable clinical accuracy, unlike rectal cancer where MRI rectal and flexible sigmoidoscopy have been well established. Importantly, conventional scans used for colon cancer staging lack precise accuracy, which may introduce the risk of overtreatment for patients with MSI-H colon cancer. On the other hand, for patients with MSI-H disease GEJ and gastric cancers where organ preservation can improve quality of life, nonoperative management with watch-and-wait approaches can be further investigated once larger studies and more evidence emerge in the era of immunotherapy. Notably, neoadjuvant immunotherapy may induce the early establishment of immunologic memory; a hypothesis suggests that MANA-primed T cells in the tumor microenvironment may increase the efficacy of immune checkpoint inhibitors to eradicate micrometastases that may otherwise trigger relapse [74]; see Figure 1. However, several other studies indicated that MANA-primed T cells can be captured in blood and lymphatic circulation, indicating that antitumor immune response is not a local but rather a systemic response [75,76,77]. Further studies are needed to better understand the local and systemic dynamics of anticancer immune response for patients undergoing ICI therapy.

Patients with MSI-H CRC with peritoneal metastasis alone appear to achieve durable responses, and cytoreductive surgeries have revealed a high incidence of complete pathological responses. Given the depth of the response, it will be an important question to determine whether these patients would need HIPEC, which is associated with several toxicities. Although cytoreductive surgery without HIPEC can provide further data to understand the biology of MSI-H CRC with peritoneal carcinomatosis, future studies are needed to understand whether the watch-and-wait approach can be considered for these patients as well. Similarly, the role of the watch-and-wait approach for patients with oligometastatic disease can be further defined with prospective studies in which the depth of response to ICI therapy for liver and lung metastasis of MSI-H CRC can be measured. However, it is important to note that growing evidence suggests that liver metastasis of MSI-H CRC is associated with inferior response compared with other sites of metastasis, and this should be considered in decision-making processes [78,79,80].

Previous studies showed that neoadjuvant ICI therapy is safe and tolerable for patients with gastrointestinal cancers [46,67]. Most of the immune-related adverse events were of grades 1 and 2. The most common immune-related adverse events of ICI therapies were skin disease (44–68%), followed by gastrointestinal reactions (5–50%) [81]. Most notably, IRAEs appear to be less common when used early in the course of the disease. For example, no grade 4 or higher toxicities were noted in dostarlimab neoadjuvant therapy for patients with locally advanced rectal cancer, while more toxicities were noted when it was used in a chemorefractory setting. Therefore, based on cumulative evidence indicating deeper and more durable responses with the earlier use of ICIs and relatively lower incidence of immune-related toxicities, these agents offer promise as future neoadjuvant therapies for MSI-H gastrointestinal cancers.

While ICI therapy offers highly promising outcomes, more research is needed to identify better biomarkers for the prediction of treatment response. Recent studies investigated surrogates of ICI response and showed that the immune signature of T and B cells may determine outcomes of ICI therapy. For example, in a multiomic study, effector T cells showed that T-cell repertoire and their affinity to neoantigens are highly linked to the effectiveness of ICI therapy [82]. In this study, patients with immune hot tumors were more likely to have reactive T cells, while other cancers, such as glioblastoma, had more hypofunctional T cells and were less likely to respond to ICI therapy alone [82]. Notably, several studies indicate there is continuous interaction between cancer cells and tumor stroma that may play an important role for the remodeling of the tumor microenvironment, which can impact antitumor immune response [83,84]. In a study of 19 patients with MSI-H CRC, single-cell RNA sequencing analysis showed that patients who achieved complete pathological response had more CD8+ T cells and CD20+ B cells in the tumor microenvironment, while those with persistent tumors had more CD4+ T regulatory cells [84]. Similar observations were noted for patients with MSI-H CRC and liver metastasis treated with ICI therapy [85]. The authors reported evidence of increased CD8+ and CD4+ activation in posttreatment samples, and, more importantly, the presence of B cells’ transcriptomic signature in the tumor microenvironment was associated with improved survival outcomes [85]. Notably, tumor mutation burden also seems to be an important marker for increased MANA load, and thus enhances clonal expansion of MANA-specific primed T cells [86].

Collectively, larger prospective clinical trials with more translation science and longer-term follow-ups are warranted to be able to draw more precise conclusions. At this time, the path for organ preservation for rectal cancer is in alignment with efforts to bring immunotherapy to a neoadjuvant setting to derive deep and durable responses to maximize organ preservation. The findings from the current literature indicate that various ICIs utilized in the neoadjuvant treatment of dMMR/MSI-H gastrointestinal cancers exhibit a high rate of complete CR, and whether organ preservation for GEJ and gastric cancers can be set as a primary endpoint of future studies warrants more studies. The role of neoadjuvant immunotherapy in MSI-H colon cancer is yet to be determined, as organ perseveration for colon cancer is clinically less relevant and may have a limited impact on the quality of life of patients with colon cancer. It is also unclear whether perioperative immunotherapy would lead to better disease-free survival outcomes compared with adjuvant settings.

## 7. Conclusions

ICI therapy has changed the treatment paradigm of patients with MSI-H gastrointestinal cancers. The depth of pathological response is highly promising and encourages further efforts to bring these highly effective agents to neoadjuvant settings. For future studies, it is critically important to define primary endpoints of prospective clinical trials when evaluating ICI-based therapies as neoadjuvant treatments. This is particularly important as the implications of each disease management strategy may have a distinct impact on the quality of life of patients, where primary endpoints should align with the expectations of patients while moving the field forward.

## Figures and Tables

**Figure 1 cancers-15-03833-f001:**
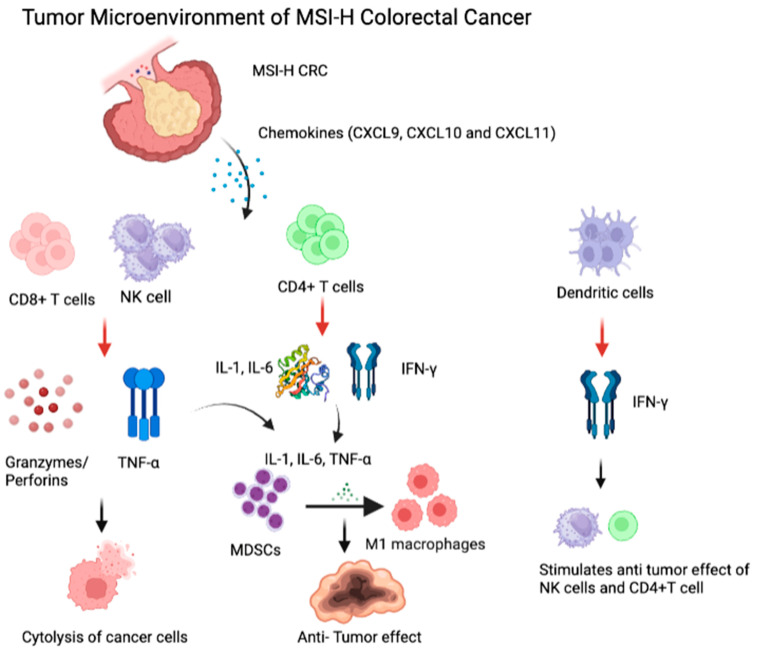
MSI-H Colorectal Cancer Tumor Microenvironment: Released chemokines (CXCL9, CXCL10, and CXCL11) recruit and activate mutation-associated neoantigen (MANA)-specific cytotoxic T cells, NK cells, and dendritic cells. CD8+ TILs cells and NK cells secrete TNF-α and granzymes/perforins, leading to the cytolytic effect of tumor cells. CD4+ TILs secrete IL-1, IL-6, and IFN-γ and dendritic cells secrete IFN-γ. The release of IL-1, IL-6, and TNF-α promotes the formation of M1 macrophages, leading to antitumor effect. Secreted IFN-γ stimulates NK cells and CD4+ TILs, leading to antitumor effect.

## Data Availability

Not applicable.

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
