# Peer review of "Neoadjuvant Immunotherapy for Patients with dMMR/MSI-High Gastrointestinal Cancers: A Changing Paradigm"

_cancers, 2023, doi:10.3390/cancers15153833_

Round 1

Reviewer 1 Report

In this paper, the authors nicely summarized a changing paradigm related to neoadjuvant immunotherapy for patients with dMMR/MSI-H GI cancers.  Congratulations!

The paper is well written and very comprehensive covering the existing literature and ongoing trials that are exploring the role of neoadjuvant IO in GI malignancies.   

Author Response

Thank you very much for the kind comments and endorsing our manuscript for publications! 

Reviewer 2 Report

In this review article, the authors provide an overview of recent advancements in neoad-juvant immunotherapy for patients with dMMR/MSI-H gastrointestinal cancers, they also elaborate on the advantages and challenges of neoadjuvant ICI therapy.

This is a comprehensive review, giving a great overview of the available literature. The topic of the review is suitable for this special issue and is of interest for the readership of this journal. The manuscript has been written well, the flow of ideas information and ideas are acceptable. The manuscript have scientific merit for publication. Therefore, I would like to suggest to accept in the present form.

Author Response

Thanks for your kind and thoughtful comments. We appreciate your endorsement for acceptance of our manuscript for publication. 

Reviewer 3 Report

This review discusses the use of immune checkpoint inhibitors (ICI) in clinical studies for the treatment of gastrointestinal cancers with a mismatch repair deficiency (MMR-D)/microsatellite instability-high (MSI-H). The use of ICIs has shown great promise with a strong impact on current cancer treatment. The review is well-written and easy to understand. One suggestion for improvement would be for the authors to briefly summarize the differences in the immune microenvironment, such as T cell infiltration, between MSI-H and MSS CRC. Additionally, it would be beneficial to include a discussion on the physiological and metabolic changes of T-cells in response to ICI. Some more recent publications (e.g., 10.1016/j.celrep.2022.111929; 10.1126/scitranslmed.add1016; 10.1016/j.ccell.2023.04.011; 10.1158/1078-0432.CCR-21-0163) could provide useful insights.

Author Response

Thanks so much for this insightful comments. We have now added another paragraph to discussion to address importance of the tumor microenvironment and discussed recent multi-omic studies suggested by the Reviewer and cited in the article. We thank so much again for this important feedback.

Round 2

Reviewer 3 Report

Excellent work.